# Seven New Species of *Entoloma* Subgenus *Cubospora* (Entolomataceae, Agaricales) from Subtropical Regions of China

**DOI:** 10.3390/jof10080594

**Published:** 2024-08-22

**Authors:** Lin-Gen Chen, Ling Ding, Hong Chen, Hui Zeng, Zhi-Heng Zeng, Sheng-Nan Wang, Jun-Qing Yan

**Affiliations:** 1Jiangxi Key Laboratory for Excavation and Utilization of Agricultural Microorganisms, Jiangxi Agricultural University, Nanchang 330045, China; chenlingen90@gmail.com (L.-G.C.); dl089311@163.com (L.D.); chenhong1238@126.com (H.C.); 2Institute of Edible mushroom, Fujian Academy of Agricultural Sciences, Fuzhou 350011, China; zenghui69@gmail.com (H.Z.); zengzhiheng@faas.cn (Z.-H.Z.); 3Jiangxi Key Laboratory of Subtropical Forest Resourecs cultivation, Jiangxi Agricultural University, Nanchang 330045, China

**Keywords:** basidiomycetes, phylogeny, seven new taxa, taxonomy

## Abstract

*Entoloma* is a relatively large genus in Agaricales, with a rich diversity of species and a wide distribution. In this study, seven new species of *Entoloma* belonging to the subgenus *Cubospora* have been identified based on morphological and phylogenetic evidence from subtropical regions of China. Morphologically, *E. excavatum* is characterized by the yellow, depressed, estriate pileus and medium-sized basidiospores; *E. lacticolor* is recognized by the white and papillate pileus, adnexed lamellae, and presence of clamp connections; *E. phlebophyllum* is identified by the pink-to-maroon and estriate pileus, and lamellae with lateral veins; *E. rufomarginatum* differs from other cuboid-spored species by the lamellae edge which is red-brown-underlined; *E. subcycneum* is characterized by the white pileus and carneogriseum-type cheilocystidia; *E. submurrayi* is recognized by the pileus margin exceeding the lamellae, 2-layered pileipellis with hyphae of different widths, and the presence of clamp connections; *E. tomentosum* is identified by the tomentose pileus, heterogeneous lamella edge, and versiform cheilocystidia with brown-yellow contents. Their distinct taxonomic status is confirmed by the positions of the seven new species in both the ITS + LSU and 3-locus (LSU, *tef-1α*, *rpb2*) phylogenetic trees. Detailed descriptions, color photos, and a key to related species are presented.

## 1. Introduction

*Entoloma* (Fr.) P. Kumm., with *E. sinuatum* (Bull.) P. Kumm designated as the type species, was established by Kummer in 1871 [1]. It is characterized by the pink spore prints and angular basidiospores viewed from any side [2]. The species of *Entoloma* are worldwide, encompassing habitats from the frigid zone to the tropics, alpine to basins, with the majority being saprobic on shady and humid ground or mosses in the forests [3,4]. So far, approximately 1800 species of *Entoloma* have been reported in the world [5,6] (https://www.speciesfungorum.org (accessed on 27 May 2024)).

In the *Entoloma* sensu lato, some species with cuboid basidiospores (with six, more or less equal, quadrangular faces, and a dihedral base) are notable due to the unique basidiospores morphology. More than 170 such species are reported worldwide, mainly occurring in the tropical and subtropical regions [7]. Traditionally, these species were classified in the subgenus *Nolanea* section *Staurospori* [4] and subgenus *Inocephalus* section *Staurospora* [8]. To clarify the taxonomic position of cuboid-spored species within *Entoloma*, analyses of phylogeny and basidiospore morphology were performed by Arstedt et al. [9]. It was found that these species formed two well-supported branches and fall into two separate subgenera: *Cubospora* Karstedt, Capelari, Largent, T.J. Baroni and Bergemann with type species *E. luteolamellatum* (Largent and Aime) Blanco-Dios and *Cuboeccilia* Karstedt, Capelari, and Largent with type species *E. omphalinoides* (Largent) Blanco-Dios [9].

In China, approximately 150 species of the *Entoloma* have been reported, with about 20 of these featuring cuboid spores, and 8 of these have been published as new species [7]. During our investigation for *Entoloma* in subtropical regions of China, some species of *Entoloma* with cuboid basidiospores were found, but do not match the known species. Based on morphological comparisons and phylogenetic analysis, some unknown-to-science cuboid-spored species were further confirmed, with seven new taxa recorded in this paper.

## 2. Materials and Methods

### 2.1. Morphological Studies

Specimens were collected from Fujian, Jiangxi, and Zhejiang provinces of China between 2020 and 2022, and were deposited in the Herbarium of Fungi, Jiangxi Agricultural University (HFJAU). Macroscopic characteristics were recorded from fresh specimens. The color codes referred to the Methuen Handbook of Colour [10]. Micromorphological structures were observed and measured under an Olympus BX53 microscope (Olympus corporation, Tokyo, Japan) by making squash preparations of sections of dried specimens that were placed in 5% KOH solution or H_2_O, and 1% Congo red was used as the staining agent when observing colorless tissues. Melzer’s reagent was selected for determining whether the spores were amyloid or not [11]. At least 20 basidiospores, basidia, and cystidia were measured for each collection. The range of spore size is expressed as the form (a) b–c (d), in which “a” and “d” representing the minimum and maximum values, and 90% of the spores falling within the range ‘b–c’. The meanings of the other spore characteristics are as follows: “Q” stands for the ratio of length and width; “av” symbolizes average value; “n” means number; and “Q_m_” indicates average “Q” ± standard deviation [12]. The morphological description is based on the work of Noordeloos et al. [13].

### 2.2. DNA Extraction, PCR Amplification, and Sequencing

Genomic DNA was extracted from dried specimens with the NuClean Plant Genomic DNA kit (CWBIO, China) [14]. The ITS, LSU, *tef-1α*, and *rpb2* regions were amplified respectively using the primer pairs of ITS1F/ITS4, LR0R/LR5 [15], EF983F/EF1953R [2], and *rpb2*-i6f-RhoF1/*rpb2*-RhoR1 [16].

PCR amplification was conducted with a 25 μL reaction system as follows: 1 µL DNA, 2 µL primers, 9.5 µL ddH_2_O, and 12.5 µL 2 × Taq Master Mix (Dye Plus). For ITS, PCR was carried out using a touchdown amplification procedure: initial 95 °C for 5 min, and then 14 cycles of denaturing at 95 °C for 30 s, annealing at 65 °C for 45 s (−1 °C per cycle), extension at 72 °C for 1 min, and then 30 cycles of denaturing at 95 °C for 30 s, annealing at 52 °C for 30 s, and extension at 72 °C for 1 min, with the final extension at 72 °C for 10 min [17]. For others, the procedure was initial 98 °C for 5 min, and then 8 cycles of denaturing at 98 °C for 5 s, annealing at 61 °C for 40 s (−1 °C per cycle), extension at 72 °C for 2 min, and then 35 cycles of denaturing at 98 °C for 5 s, annealing at 54 °C for 1.5 min, extension at 72 °C for 2 min, with the final extension at 72 °C for 10 min. The PCR products were sequenced by Qing Ke Biotechnology Co. Ltd. (Wuhan City, China).

### 2.3. Alignment and Phylogenetic Analyses

Since the ITS data are absent for many species of *E.* subgenus *Cubospora*, two phylogenetic trees based on ITS + LSU and LSU + *tef-1α* + *rpb2*, were constructed by Bayesian inference (BI) and Maximum likelihood (ML), respectively, according to the previous studies of Karstedt et al. [9] and Morozova and Pham [18]. Some species of *E.* subgenus *Nolanea* are designated as outgroups. Information on specimens and GenBank accession numbers are listed in Table 1. ITS, LSU, *tef-1α*, and *rpb2* sequence datasets were separately aligned on the MAFFT online server [19]. BI and ML phylogenetic analyses of the processed sequences were run using Mrbayes v.3.2.7a and IQtree v.2.1.2, respectively [20]. The best-fit models of ML and BI were determined by PartitionFinder [21] complying with Corrected Akaike information criterion (AICc). For the ML analysis, 1000 replicates are performed based on the ultrafast bootstrap option of ML that allowed partitions from different seeds. For the BI analysis, the genes chains were run for 2,000,000 generations. The first 25% of trees were discarded as burn-in. The branches of Bayesian posterior probability (BI-PP) ≥ 0.95 and ML bootstrap support (ML-BP) ≥ 75% are considered as statistical supports. The identifying criteria for new species are, according to the viewpoints proposed by Dettman et al., exhibiting 1 to 2 stable morphological differences from similar species and forming separated and stable clades in the phylogenetic tree [22].

## 3. Results

### 3.1. Phylogenetic Analysis

In total, 243 sequences of 96 samples were used for phylogenetic analyses. For the IL (ITS + LSU) tree, a total of 2135 characters were used in the analyses, of which 1494 were constant, 496 were parsimony-informative, and 145 were singleton. The best-fit models of both ML and BI were the same: GTR + F + I + G4 for ITS and HKY + F + I + G4 for LSU. For the LTR (LSU + *tef-1α* + *rpb2*) tree, a total of 3033 characters were used in the analyses, of which 1998 were constant, 809 were parsimony-informative, and 226 were singleton. The best-fit models of both ML and BI were the same: GTR + F + I + G4 for LSU and *tef-1α*, SYM + I + G4 for *rpb2*. For Bayes analysis, the average standard deviation of split frequencies is less than 0.01 after 650,000 generations.

The results of the phylogenetic analysis are shown in the Figure 1 (IL tree) and Figure 2 (LTR tree). The overall frameworks were consistent with previous studies [9]. Seven new species were all clustered in the subgenus *Cubospora* clade and formed separated and stable branches, respectively. In the IL tree, *E. excavatum*, *E. lacticolor*, and *E. rufomarginatum* formed distinct lineages, separately (BI-PP = 1, ML-BP = 100%). *E. submurrayi* formed a sister lineage with *E. murrayi* (Berk. and M.A. Curtis) Sacc. and P. Syd. (BI-PP = 1, ML-BP = 100%). *E. phlebophyllum* formed a sister lineage with *E. tomentosum* (BI-PP = 1, ML-BP > 95%), and *E. subcycneum* formed a sister lineage with *E. cycneum* O.V. Morozova and T.H.G. Pham (BI-PP = 1, ML-BP = 100%). In the LTR tree, *E. lacticolor*, *E. phlebophyllum*, *E. subcycneum*, and *E. tomentosum* formed distinct lineages, separately (BI-PP > 0.95, ML-BP ≥ 75%). *E. subcycneum* formed a sister lineage with *E. cycneum* and *E. peristerinum* O.V. Morozova and T.H.G. Pham (BI-PP = 1, ML-BP = 100%).

### 3.2. Taxonomy

***Entoloma excavatum*** J.Q. Yan, L.G. Chen, and S.N. Wang sp. nov. (Figure 3A and Figure 5).

MycoBank: MB854072

**Figure 3 jof-10-00594-f003:**
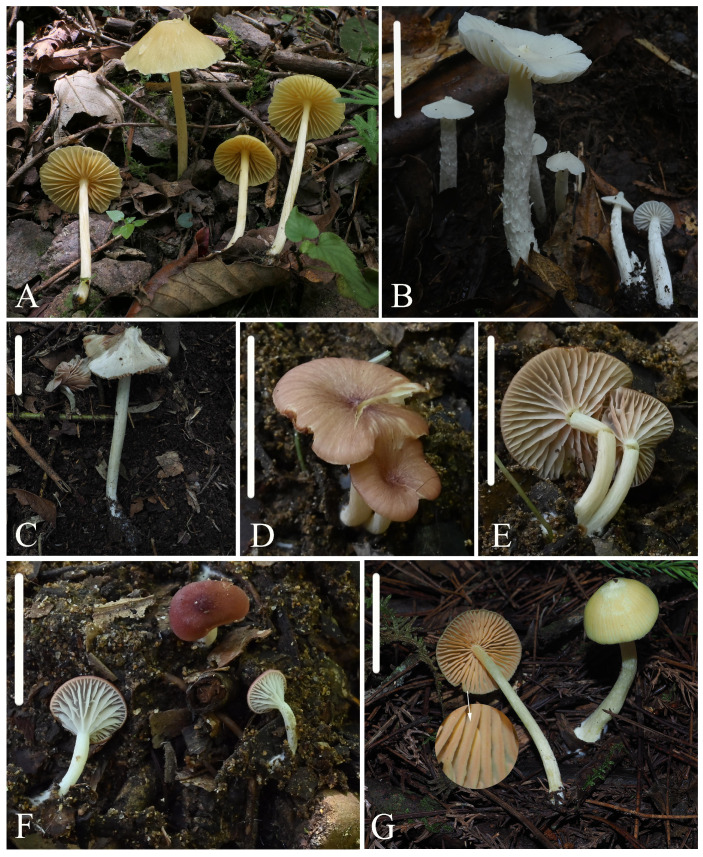
Photos of basidiomata. (**A**) *Entoloma excavatum*: HFJAU2013, holotype; (**B**,**C**) *Entoloma lacticolor*: (**B**) HFJAU3064, (**C**) HFJAU3736, holotype; (**D**–**F**) *Entoloma phlebophyllum*: (**D**,**E**) HFJAU4261, holotype; (**F**) HFJAU4263; (**G**) *Entoloma rufomarginatum*: HFJAU1933, holotype. Scale bars: (**A**,**D**–**G**) 20 mm; (**B**,**C**) 30 mm.

**Figure 4 jof-10-00594-f004:**
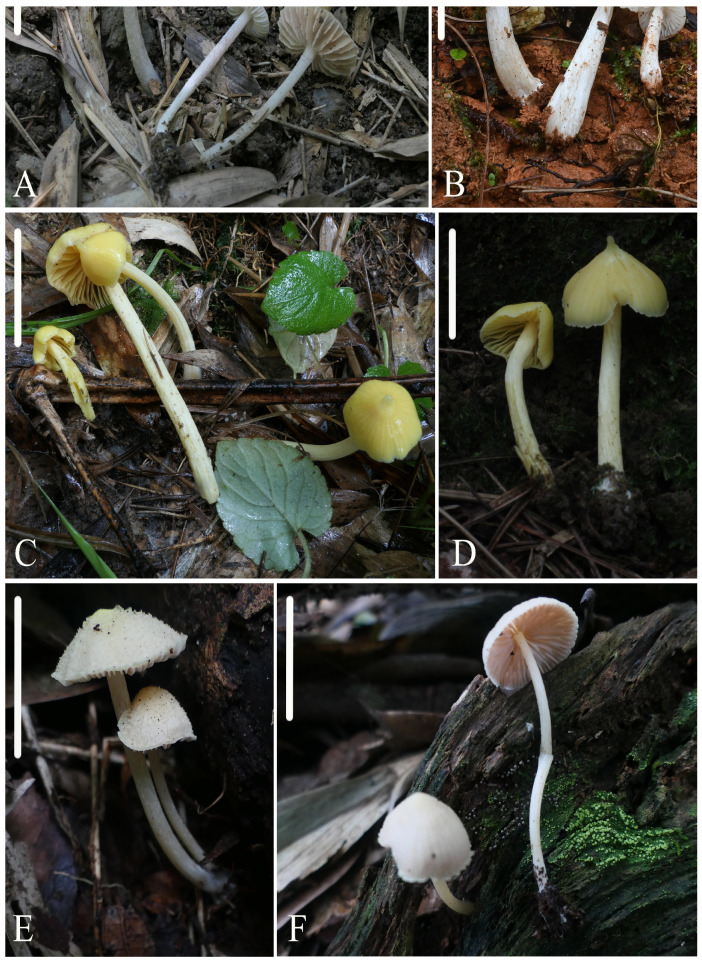
Photos of basidiomata. (**A**,**B**) *Entoloma subcycneum*: (**A**) HFJAU0985, (**B**) HFJAU3124, holotype; (**C**,**D**) *Entoloma submurrayi*: (**C**) HFJAU1050, (**D**) HFJAU3587, holotype; (**E**,**F**) *Entoloma tomentosum*: (**E**) HFJAU5159, holotype, (**F**) HFJAU5166. Scale bars: (**A**–**F**) 20 mm.

*Etymology*. Refer to Latin “excavatus” (excavate)—the pileus is excavated.

*Holotype*. China, Zhejiang Province, Lishui City, Jiulongshan Reserve in Suichang County, 28°23′23″ N, 118°51′0″ E, alt. 535 m, 14 July 2020, collected by Yu-Peng Ge, Bin-Ron Ke, and Zhi-Heng Zeng, HFJAU2013.

*Diagnosis*. *Entoloma excavatum* is mainly characterized by the rather small and mycenoid basidiomata; conical to broadly conical with a depression at center; bright yellow, estriate, and glabrous pileus; relatively sparse, adnexed, and subventricose lamellae with crenulate edge; glabrous and striae stipe; cuboid basidiospores; sterile lamellae edge; cylindrical cheilocystidia; presence of clamp connections. It differs from *E. overeemii* E. Horak by its mycenoid basidiomata, darker estriate pileus, adnexed lamellae, larger basidiospores, cylindrical cheilocystidia, and presence of clamp connections.

*Description*. *Basidiomata,* rather small, mycenoid. *Pileus* 13–22 mm wide, conical to broadly conical with a depression at the center, glabrous or sparsely fibrillous, not translucently striate, with entire or serrate margin, bright yellow (3C7–8). *Lamellae* relatively sparse, 1.5–2.0 mm wide, with two types of lamellules, adnexed to sinuate, ventricose, concolored with cap, with serrate and concolorous edge. *Stipe* 30–50 × 2.0–3.0 mm, central, terete, hollow, equal, concolorous with the pileus, slightly with longitudinal or oblique striae, smooth and glabrous, base with white tomentum. *Context* is thin, concolorous to the surface. *Odor* is indistinct, *taste* is not tested.

*Basidiospores* are (8.0) 8.5–10.5 × 8.0–10.0 (10.5) μm, (av = 9.3 × 8.8 μm), Q = 1.0–1.1 (1.2) (Q_m_ = 1.05 ± 0.03, n = 60), isodiameterical, cuboid, sporadically with five angles in side-view, thick-walled, inamyloid. *Basidia* are 40–52 × 11–13 μm, clavate, 4-spored, sterigmata 8.0–11 μm long, clamped. *Pleurocystidia* is absent. *Cheilocystidia* are 50–96 × 8.0–13 μm, serrulatum-type, irregular clusters in the sterile lamellae edge, cylindrical, with rounded apex. *Lamellar trama* are regular, made up of cylindrical hyphae 5.0–13 µm wide, with oleiferous hyphae near to the margin. *Pileipellis* is a cutis with transitions to a trichoderm towards the margin, made up of cylindrical hyphae 5.0–11 μm broad, thin-walled, even at septa, with rounded end and sparsely pale yellow encrusting pigment, easily dissolved in KOH solution. *Stipitipellis* is a cutis composed of cylindrical hyphae 5.0–9.0 μm wide, slightly constricted at septa. *Clamp connections* are present and abundant in all parts of the basidiocarp.

*Habitat*. Scattered on soil in mixed coniferous-broad-leaved forest.

Distribution. China.

*Additional specimens examined*: China, Zhejiang Province, Lishui City, Jiulongshan Reserve in Suichang County, 28°23′23″ N, 118°51′2″ E, alt. 496 m, 14 July 2020, collected by Yu-Peng Ge, Bin-Ron Ke, and Zhi-Heng Zeng, HFJAU4774.

*Notes*. In the IL trees, this specimen *E. murrayi* MHHNU 30,602 should be *E. excavatum,* since it shares 99.3% similarity with our species in ITS and clusters into a stable branch (BI-PP = 1, ML-BP = 100%). Additionally, *E. murrayi* and *E. quadratum* (Berk. and M.A. Curtis) E. Horak are closely related to the new species. *E. murrayi* differs from *E. excavatum* by the striate pileus without a depression at the center and a lack of clamp connections [38]. *E. quadratum* is distinguished by its orange-yellow to salmon and striate pileus without a depression at the center [38]. Morphologically, *E. excavatum* shares many features with *E. overeemii*, including the yellow, depressed umbilicate, glabrous pileus and the cuboid basidiospores. However, *E. overeemii* is distinguished by the omphaloid basidiomata, translucently striate pileus, adnate to decurent lamellae, smaller basidiospores (5.0–7.0 μm), clavate to vesiculose cheilocystidia, and lack of clamp connections [38].

**Figure 5 jof-10-00594-f005:**
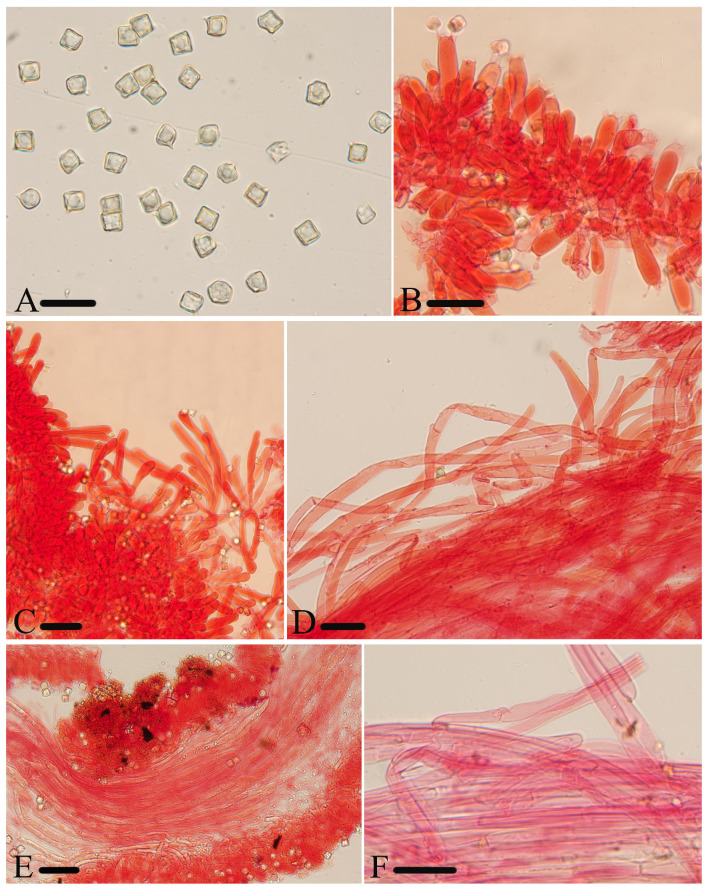
Micromorphological structures of *Entoloma excavatum.* (**A**) Basidiospores. (**B**) Basidia. (**C**) Cheilocystidia. (**D**) Pileipellis. (**E**) Lamellar trama. (**F**) Stipitipellis. Scale bars: (**A**–**F**) 30 μm. All structures were observed in 5% KOH, and 1% Congo red was used as the stain, except (**A**).

***Entoloma lacticolor*** J.Q. Yan, L.G. Chen, and S.N. Wang, sp. nov. (Figure 3B,C and Figure 6).

MycoBank: MB850724

*Etymology*. Refer to Latin “lacteus” (milky white)—the basidiomata is entirely milk-white.

*Holotype*. China, Fujian Province, Wuyishan City, Zhongpeng Village, 27°55′18″ N, 117°51′13″ E, alt. 674 m, 25 June 2022, collected by Yu-Peng Ge and Meng-Hui Han, HFJAU3736.

*Diagnosis*. *Entoloma lacticolor* is mainly characterized by the milky white and glabrous pileus with obvious acute papilla; serrate lamellae edge; glabrous or furfuraceous-scaly stipe with longitudinal or oblique groove; cuboid basidiospores; presence of clamp connections. It differs from *E. album* Hiroë by the cylindrical to subclavate cheilocystidia and presence of clamp connections.

*Description*. *Basidiomata* are small to medium-sized. *Pileus* is 15–50 mm wide, conical when young, from hemispherical to flattened with age, with obvious acute papilla at the center, not hygrophanous, smooth, glabrous, marked by translucently radial striae almost up to 2/3 of the radius, with slightly cracking, straight, glabrous or squamulose margin, milky white (1A1–3A1), yellowish at center. *Lamellae* are medium density, 2.0–8.0 mm wide, with two types of lamellules, adnexed, ventricose, originally white, becoming pink (9A3), with a serrulate and concolourous edge. *Stipe* are 22–110 × 2.5–15 mm, central, cylindrical, hollow, equal, concolored with cap, with longitudinal or oblique groove, glabrous or furfuraceous-scaly, base with white tomentum. *Smell and taste* are indistinct.

*Basidiospores* are (7.5) 8.0–10.0 (10.5) × (7.0) 7.5–9.0 (10.0) μm, (av = 8.7 × 8.2 μm), Q = 1.0–1.1 (1.2) (Q_m_ = 1.05 ± 0.04, n = 80), isodiameterical, cuboid, sporadically with five angles in side-view, thick-walled, inamyloid. *Basidia* are 35–53 × 10–14 μm, clavate, 4-spored, sterigmata 6.0–14 μm long, clamped. *Pleurocystidia* are absent. *Cheilocystidia* are 33–75 × 7.0–14 μm, carneogriseum-type, dispersed along the sterile lamellae edge, cylindrical to subclavate, septate, with a rounded apex. *Lamellar trama* is regular, made up of cylindrical hyphae 9.0–15 µm wide, intertwined by refractive hyphae near the margin. *Pileipellis* is a cutis to a trichoderm of cylindrical hyphae 5.0–11 μm broad, thin-walled, slightly constricted at septa, without pigment. *Stipitipellis* is a cutis composed of cylindrical hyphae, up to 13 μm wide, slightly constricted at septa. *Clamp-connections* are present in all tissue.

*Habitat*. Scattered or solitary on soil in mixed coniferous-broad-leaved forest.

Distribution. China

*Additional specimens examined*. China, Fujian Province, Wuyishan City, 27°50′9″ N, 117°46′25″ E, alt. 1905 m, 11 August 2021, collected by Jun-Qing Yan and Ze-Wei Liu, HFJAU1392, HFJAU1393, HFJAU3064; Wuyishan City, Zhongpeng Village, 27°55′15″ N, 117°51′12″ E, alt. 695 m, 25 June 2022, collected by Yu-Peng Ge, and Meng-Hui Han, HFJAU3721, HFJAU3728, HFJAU3737, HFJAU3744.

*Notes*. In the IL trees, *E. rufomarginatum* is rather close to *E. lacticolor.* But, the former differs in its yellow pileus, lamellae edge red-brown underlined, and shows only 94.8% similarity with new species in the ITS sequence. Morphologically, some species have aspects of *E. lacticolor* with cuboid basidiospores and white basidiomata, but can be separated as follows. *E. albidoquadratum* Manim. and Noordel. have pleurocystidia and larger basidiospores (11–16.5 × 9–14 μm) [39]. *E. cycneum* differs in its pileus without obviously acute papilla at the center [18]. *E. peristerinum* differs in its porphyrogriseum-type cheilocystidia [18]. *E. minutoalbum* E. Horak is a common species of New Zealand, and has smaller basidiospores (6–8 μm) [38]. *E. subcycneum* is distinguished by its smaller pileus and ITS region, with a similarity of 93.8%.

**Figure 6 jof-10-00594-f006:**
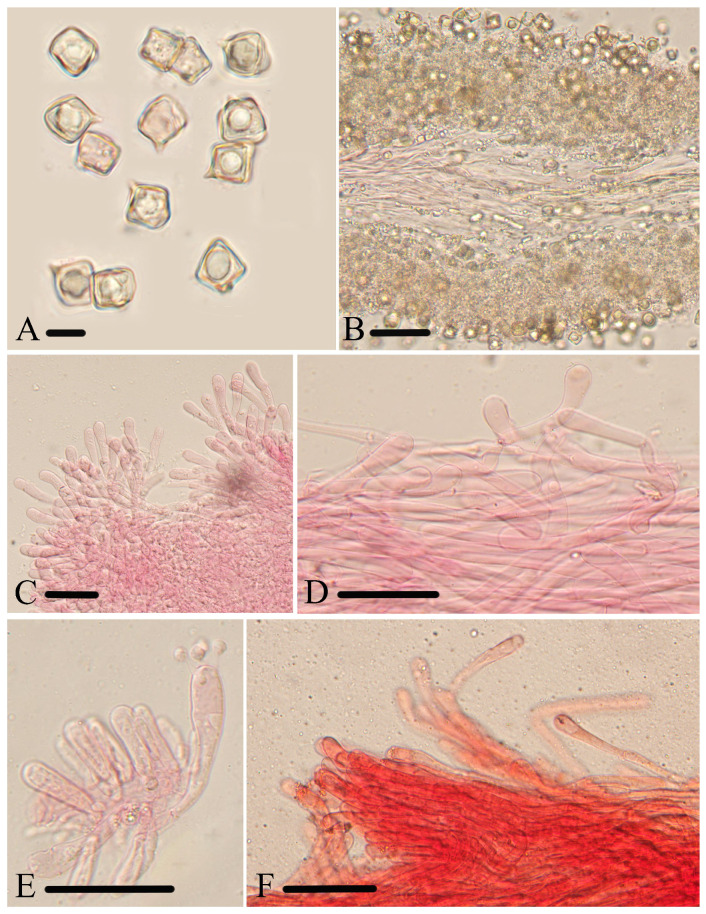
Micromorphological structures of *Entoloma lacticolor.* (**A**) Basidiospores. (**B**) Lamellar trama. (**C**) Cheilocystidia. (**D**) Stipitipellis. (**E**) Basidia. (**F**) Pileipellis. Scale bars: (**A**) 10 μm; (**B**–**F**) 30 μm. All structures were observed in 5% KOH, and 1% Congo red was used as the stain, except (**A**,**B**).

***Entoloma phlebophyllum*** J.Q. Yan, L.G. Chen, and S.N. Wang sp. nov. (Figure 3D–F and Figure 7).

MycoBank: MB851118.

*Etymology*. According to the features of lamellae—having tiny lateral veins.

**Figure 7 jof-10-00594-f007:**
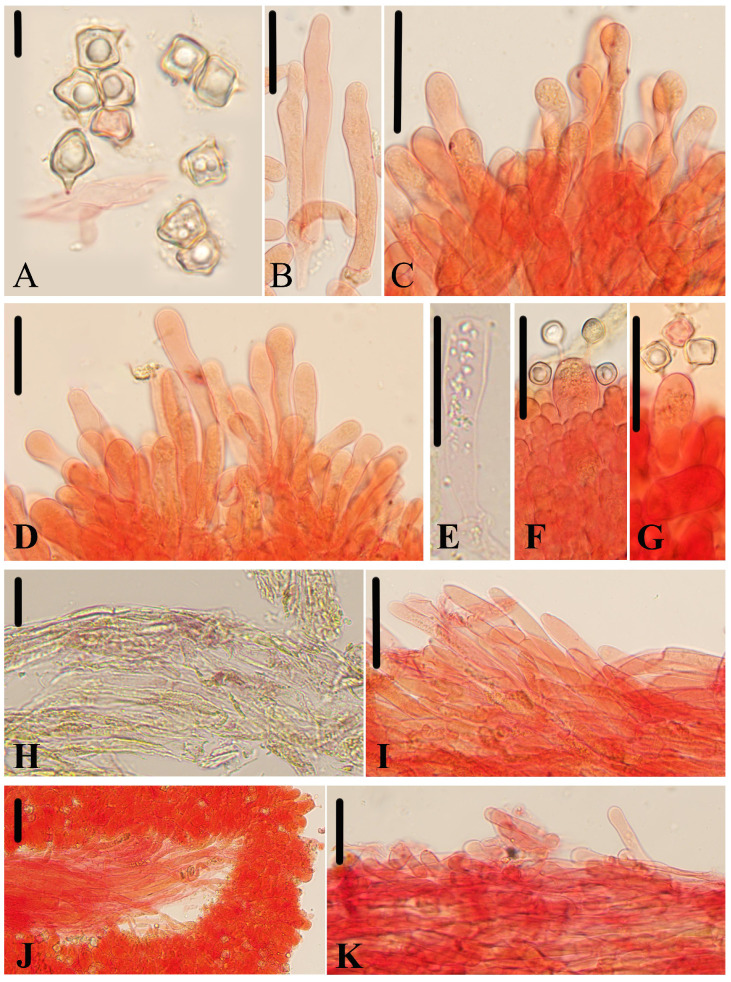
Micromorphological structures of *Entoloma phlebophyllum.* (**A**) Basidiospores. (**B**–**D**) Cheilocystidia. (**E**–**G**) Basidia. (**H**,**I**) Pileipellis. (**J**) Lamellar trama. (**K**) Stipitipellis. Scale bars: (**A**) 10 μm; (**B**–**K**) 30 μm. All structures were observed in 5% KOH, and 1% Congo red was used as the stain, except (**H**).

*Holotype*. China, Fujian Province, Tongmuguan Pass, 27°41′45″ N, 117°49′20″ E, alt. 391 m, 11 July 2022, collected by Jun-Qing Yan and Cheng-Feng Nie, HFJAU4261.

*Diagnosis*. *E. phlebophyllum* is recognized by the conical-to-plano convex with a depression, pink-to-maroon and glabrous pileus; adnate to decurrent lamellae with obviously tiny lateral veins and a serrulate edge; stipe with a different color from the cap; cuboid basidiospores; sterile lamellae edge; cylindrical or clavate-capitate cheilocystidia; and the presence of clamp connections. It differs from *E. phleboides* (Romagn.) E. Horak by the pink-to-maroon pileus, presence of cystidia, and clamp connections.

*Description*. *Basidiomata* are rather small. *Pileus* are 6.0–19 mm wide, broadly conical when young, convex-to-plano convex with age, with a depressed center, not hygrophanous and translucently striate, glabrous or tomentulose towards the entire and slightly inrolled margin, squamous at the center, with serrate or straight margin, pink (8A4–9A3), dark carneous (8A3–8B3) to maroon (10E8–10E7), and darker at the center. *Lamellae* are medium density and 2.0–3.0 mm wide, with tiny lateral veins and three types of lamellules, adnate to decurrent, originally white, becoming pink, with serrulate and a concolourous or paler edge. *Stipe* are 12–43 × 1.5–2.5 mm, central, cylindrical, hollow, equal or attenuated upwards, white to dirty white, glabrous or sparsely covered by white-fibrillous scales, somewhat longitudinally striate, and white tomentose at the base. *Context* is thin and white. *Smell and taste* are indistinct.

*Basidiospores* are (7.0) 7.5–10.0 (11.0) × (6.5) 7.0–9.0 (10.0) μm, (av = 8.5 × 8.0 μm), Q = 1.0–1.2 (1.3) (Q_m_ = 1.07 ± 0.06, n = 80), isodiameterical or subisodiameterical, cuboid, rarely 3 or 5 angles in side-view, thick-walled, and inamyloid. *Basidia* are 40–55 × 11–14 μm, clavate, 4- or 2-spored, sterigmata 5.0–11 μm long, and clamped. *Pleurocystidia* are absent. *Cheilocystidia* are 37–73 × 6.0–12 μm, serrulatum-type, irregular clusters in the sterile lamellae edge, cylindrical, clavate, septate, often with rounded apex, and sometimes with tapered apex or constricted neck. *Lamellar trama* are regular, made up of cylindrical hyphae 4.0–10 µm wide, and intertwined by oleiferous hyphae near the margin. *Pileipellis* is a cutis to a trichoderm of cylindrical hyphae 5.0–13 μm broad, thin-walled, slightly constricted at septa, and with a tapered or pointed end and brownish yellow encrusting pigment and brick red intracellular pigment. *Stipitipellis* is a cutis composed of densely arranged, cylindrical hyphae, up to 10 μm wide, and slightly constricted at the septa. *Brilliant granules* are abundant and *clamp connections* are present in all tissue.

*Habitat*. Scattered on soil in mixed coniferous-broad-leaved forest.

Distribution. China.

*Additional specimens examined*. China, Fujian Province, Wuyi Mountain, 27°42′37″ N, 117°51′30″ E, alt. 530 m, 13 August 2021, collected by Jun-Qing Yan and Ze-Wei Liu, HFJAU3126; Fujian Province, Tongmuguan Pass, 27°41′48″ N, 117°49′19″ E, alt. 394 m, collected by Jun-Qing Yan and Cheng-Feng Nie, HFJAU4263.

*Notes*. *E. phlebophyllum* groups together with *E. luteum* Peck in the IL tree, and groups together with *E. carneum* Z.S. Bi, *E. pallidoflavum* (Henn. and E. Nyman) E. Horak, and *E. plicatum* (Largent) in the LTR tree. However, the last four species do not have tiny lateral veins between the lamellae. Apart from that, *E. carneum* differs in the striate pileus and shows only 98.5% similarity with *E. phlebophyllum* in *tef-1α* and *rpb2* sequence, respectively [40,41]; *E. luteum* differs in its yellow and striate pileus, adnexed lamellae with fimbriate edge [38], and shows only 95.8% similarity with *E. phlebophyllum* in ITS sequence; *E. pallidoflavum* differs in the pale yellow and sulcate pileus [38], and shows only 95.5% similarity with *E. phlebophyllum* in ITS sequence; *E. plicatum* is distinguished by the yellow and plicate-striate pileus with mammillate umbo [34], and shows only 98% similarity with *E. phlebophyllum* in *tef-1α* and *rpb2* sequence. In addition, morphologically, *E. infundibuliforme* Petch and *E. significum* Corner and E. Horak have aspects of *E. phlebophyllum*, but *E. infundibuliforme* differs in the smaller basidiospores (5.5–8 μm), presence of fusoid pleurocystidia, and lack of clamp connections [42]. *E. significum* is distinguished from new species by its absence of cystidia [38].

***Entoloma rufomarginatum*** J.Q. Yan, L.G. Chen, and S.N. Wang, sp. nov. (Figure 3G and Figure 8).

MycoBank: MB854068

*Etymology*. “Rufo-” means “red”, “marginata” means “margin” (Latin)—refers to the feature of the lamellae edge.

*Holotype*. China, Zhejiang Province, Lishui City, Baishanzu in Qingyuan County, 27°45′26″ N, 117°11′59″ E, alt. 1586 m, 8 July 2020, collected by Yu-Peng Ge and Qin Nan, HFJAU1933.

**Figure 8 jof-10-00594-f008:**
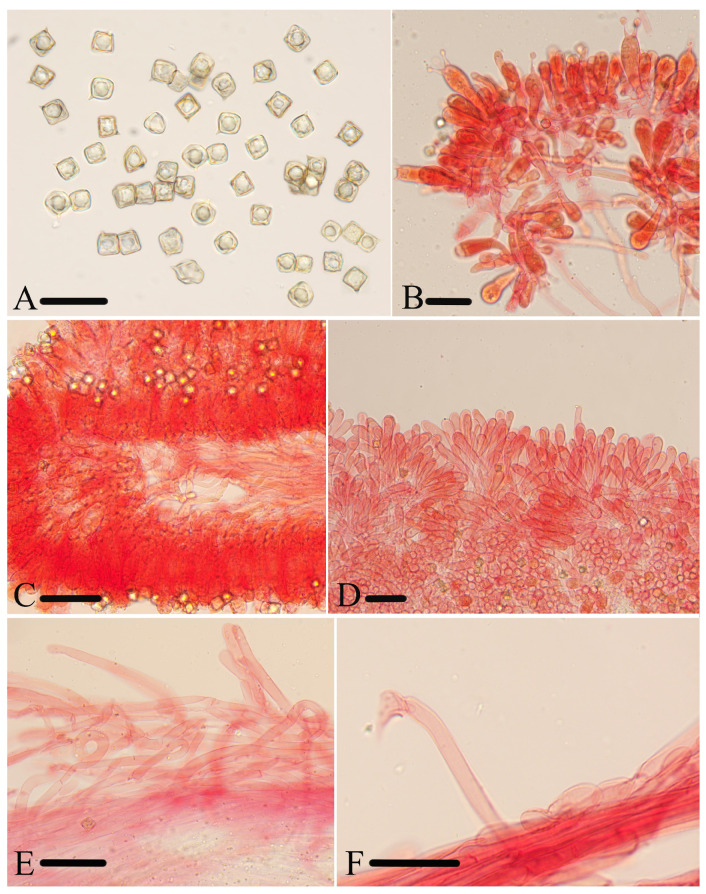
Micromorphological structures of *Entoloma rufomarginatum.* (**A**) Basidiospores. (**B**) Basidia. (**C**) Lamellar trama. (**D**) Cheilocystidia. (**E**) Pileipellis. (**F**) Stipitipellis. Scale bars: (**A**–**F**) 30 μm. All structures were observed in 5% KOH, and 1% Congo red was used as the stain, except (**A**).

*Diagnosis*. *Entoloma rufomarginatum* is mainly characterized by the rather small and mycenoid basidiomata, campanulate, or conical with acute papilla, dark brownish yellow, striae, and glabrous pileus; adnate and orange-brown lamellae with lamellae edge red-brown underlined; tomentose stipe; cuboid basidiospores; sterile lamellae edge; cylindric-clavate cheilocystidia; and presence of clamp connections. It differs from *E. gracilius* E. Horak by its lamellae edge red-brown underlined and larger basidiospores.

*Description*. *Basidiomata* are rather small, mycenoid. *Pileus* are 10–20 mm wide, campanulate or conical with obvious acute papilla, translucently striate at the margin up to the centre, slightly hygrophanous, smooth, glabrous, felted-scaly towards the entire and slightly inrolled margin, light blond (4C4–5), with green tints on margin. *Lamellae* are moderately distant, 1.5–2.0 mm wide, with two types of lamellules, adnate, emarginate, subventricose, orange-brown (6B4–5), with undulate, concolorous edge and red-brown underlined. *Stipe* are 30–55 × 4.0–8.0 mm, central, terete, hollow, equal or attenuated upwards, greenish-yellow (3CD4), tomentose, somewhat with longitudinal striae, base with white tomentum. *Context* is thin, concolorous to the surface. *Odor* is indistinct, *taste* is not tested.

*Basidiospores* are (8.0) 9.0–10.0 (12.5) × (8.0)8.5–10.0 (11.0) μm, (av = 9.8 × 9.3 μm), Q = 1.0–1.1 (1.2) (Q_m_ = 1.05 ± 0.03, n = 100), isodiameterical, cuboid, thick-walled, inamyloid. *Basidia* are 39–62 × 11–15 μm, clavate or narrowly vesiculose, 4- or 2-spored, sterigmata 7.0–14 μm long, clamped. *Pleurocystidia* are absent. *Cheilocystidia* are 35–113 × 6.0–11 μm, serrulatum-type, irregular clusters in the sterile lamellae edge, cylindric-clavate, with rounded, rarely acute apex. *Lamellar trama* are regular, made up of cylindrical hyphae 5.0–18 µm wide. *Pileipellis* is a transition between cutis and trichoderm, made up of hyphae 9.0–16 μm wide, thin-walled, even at septa, with a rounded end. *Stipitipellis* is a cutis composed of cylindrical hyphae 5.0–12 μm wide, slightly constricted at the septa, with a rounded end. *Clamp connections* are present in all tissue.

*Habitat*. Scattered on soil in mixed coniferous-broad-leaved forest.

Distribution. China.

*Additional specimens examined*. China, Zhejiang Province, Lishui City, Baishanzu in Qingyuan County, 27°45′26″ N, 119°11′59″ E, alt. 1574 m, 8 July 2020, collected by Yu-Peng Ge and Qin Nan, HFJAU4070, HFJAU4094.

*Notes*. In the IL tree, *E. lacticolor* is closest to *E. rufomarginatum*. But, the former differs in the white basidiomata, lamellae edge is not underlined, and shows only 94.8% similarity with *E. rufomarginatum* in ITS sequence. In addition, *E. avilanum* (Dennis) E. Horak, *E. kamerunense* (Bres.) E. Horak, and *E. submurrayi* are rather close to *E. rufomarginatum*, but the former three exhibit concolored lamellae with the cap and the lamellae edge not underlined. Apart from those, *E. avilanum* from Venezuela differs from *E. rufomarginatum* by the free lamellae with a subfimbriate edge [38]. *E. kamerunense* is distinguished by the absence of cystidia, and the brown plasmatic pigment in pileipellis [38].

***Entoloma subcycneum*** J.Q. Yan, L.G. Chen, and S.N. Wang sp. nov. (Figure 4A,B and Figure 9).

MycoBank: MB854069.

**Figure 9 jof-10-00594-f009:**
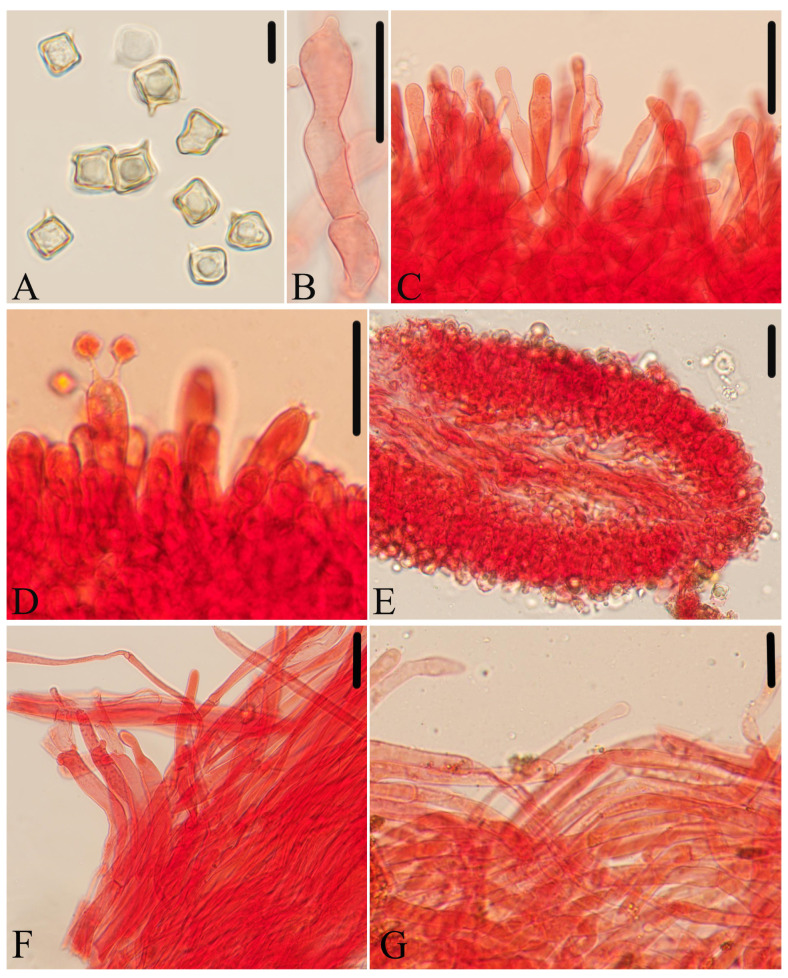
Micromorphological structures of *Entoloma subcycneum.* (**A**) Basidiospores. (**B**,**C**) Cheilocystidia. (**D**) Basidia. (**E**) Lamellar trama. (**F**) Pileipellis. (**G**) Stipitipellis. Scale bars: (**A**) 10 μm; (**B**–**G**) 30 μm. All structures were observed in 5% KOH, and 1% Congo red was used as the stain, except (**A**).

*Etymology*. Macroscopic morphology similar to “*Entoloma cycneum*”.

*Holotype*. China, Fujian Province, Wuyishan City, 27°44′43″ N, 117°41′6″ E, alt. 792 m, 13 August 2021, collected by Jun-Qing Yan and Ze-Wei Liu, HFJAU3124.

*Diagnosis*. *Entoloma subcycneum* is mainly characterized by the rather small basidiomata; conical to applanate and pure white pileus; adnexed or sinuate lamellae; cuboid basidiospores; sterile lamellae edge; cylindric-clavate and septate cheilocystidia with various apex; and the presence of clamp connections. It differs from *E. cycneum* by its carneogriseum-type cheilocystidia and absence of brilliant granules.

*Description*. *Basidiomata* are rather small. *Pileus* are 8.0–15 mm wide, conical to hemispherical when young, becoming applanate with a small depression at the center, glabrous or erected velutinous, margin entire, sometimes undulating or cracking, when moist translucently striate up to 2/3 of the radius, pure white (3A1–4A1), with a slightly pink pigment (9A3–4). *Lamellae* are moderately distant, 1.5–3.5 mm wide, with two types of lamellules, adnexed, sinuate, ventricose, white to pink, with dentate, sometimes fimbriate, and concolorous edge. *Stipe* are 15–30 × 3–6 mm, central, terete, hollow, equal or attenuated towards the apex, concolored with cap, slightly with longitudinal striae, glabrous or densely tomentous, easily peeling, and have a base with a white tomentum. *Context* is thin, white. *Odor* is indistinct, *taste* is not tested.

*Basidiospores* are (8.0) 8.5–11.0 (11.5) × (7.0) 8.0–10.0 (11.0) μm, (av = 9.5 × 9.0 μm), Q = 1.0–1.1 (1.2) (Q_m_ = 1.06 ± 0.04, n = 80), isodiameterical, cuboid, sporadically with five angles in side-view, thick-walled, inamyloid. *Basidia* are 41–54 × 11–14 μm, clavate, 4-spored, sterigmata 6.0–14 μm long, clamped. *Pleurocystidia* are absent. *Cheilocystidia* are 29–110 × 6.0–13 μm, carneogriseum-type, regularly dispersed along the sterile lamellae edge, cylindric-clavate, septate, with rounded, mucronate, acuminate apex, rarely with papilla. *Lamellar trama* are regular, made up of cylindrical hyphae 4.0–9.0 µm wide, intertwined by refractive hyphae near the margin. *Pileipellis* is a cutis with transitions to a trichoderm towards margin, made up of cylindrical hyphae 4.0–14 μm broad, thin-walled, slightly constricted at the septa, with tapered end and sparsely pale yellow encrusting pigment, easily dissolved in KOH solution. *Stipitipellis* is a cutis composed of cylindrical hyphae 4.0–8.0 μm wide. *Clamp connections* are present in all tissue.

*Habitat*. Scattered or solitary on soil in mixed coniferous-broad-leaved forest.

Distribution. China

*Additional specimens examined*. China, Jiangxi Province, Matsu Mountain, 29°37′51″ N, 116°5′25″ E, alt. 214 m, 2 July 2019, collected by Jun-Qing Yan, HFJAU0985; Fujian Province, Wuyishan City, Kuzhukeng, 27°43′52″ N, 117°51′21″ E, alt. 591 m, 25 July 2022, collected by Jun-Qing Yan and Bin-Ron Ke, HFJAU3939; Wuyi Mountain 27°44′43″ N, 117°41′6″ E, alt. 816 m, 13 August 2021, collected by Jun-Qing Yan and Ze-Wei Liu, HFJAU4738

*Notes*. In the two phylogenetic trees, *E. subcycneum* groups together with *E. cycneum* and *E. peristerinum*, but *E. cycneum* shows 95.4%, 98.3%, and 98.5% similarity with *E. subcycneum* in ITS, LSU, and *tef-1α* sequence, *E. peristerinum* shows 86.1%, 94.6%, and 94.8% similarity, respectively. In addition, *E. cycneum and E. peristerinum* are characterized by the presence of abundant brilliant granules, and having differentiated type of cheilocystidia and porphyrogriseum-type in *E. peristerinum* as well as serrulatum-type in *E. cycneum* [18].

Morphologically, *E. caribaeum* (Pegler) Courtec. and Fiard differs in the free lamellae, larger basidiospores (12–17 × 11–15 μm), 2-spored basidia, and lack of clamp connections [43]. *E. cuboidoalbum* Noordel. and Hauskn. is recognized by the typical omphaloid basidiomata, estriate pileus, forked lamellae, heterogeneous lamellae edge, and abundant cheilocystidia with a variable shape from cylindrico-clavate to lageniform or fusiform [44]. *E. galericolor* Courtec. exhibits omphaloid basidiomata, arcuate lamellae, beige brown stipe, smaller basidiospores (7–9 μm), and trichodermal pileipellis [45].

***Entoloma submurrayi*** J.Q. Yan, L.G. Chen, and S.N. Wang, sp. nov. (Figure 4C,D and Figure 10).

MycoBank: MB854070

*Etymology*. Morphology similar to “*Entoloma murrayi*”.

*Holotype*. China, Fujian Province, Nanping City, Lingxia Creek, 27°32′31″ N, 117°28′27″ E, alt. 425 m, 7 June 2022, collected by Jun-Qing Yan and Lin-Gen Chen, HFJAU3587.

*Diagnosis*: *Entoloma submurrayi* is mainly characterized by the small and mycenoid basidiomata; conical to campanulate with acute papilla, yellow, glabrous, and striae pileus; pileus margin exceeding the lamellae; adnexed lamellae; glabrous and striae stipe; cuboid basidiospores; sterile lamellae edge; cylindric-clavate cheilocystidia; 2-layered pileipellis made up of suprapellis and thinner subpellis of oleiferous hyphae; and the presence of clamp connections. It differs from *E. flavoquadratum* C.K. Pradeep and K.B. Vrinda by its mycenoid basidiomata and translucently striate pileus with obvious acute papilla at the center.

**Figure 10 jof-10-00594-f010:**
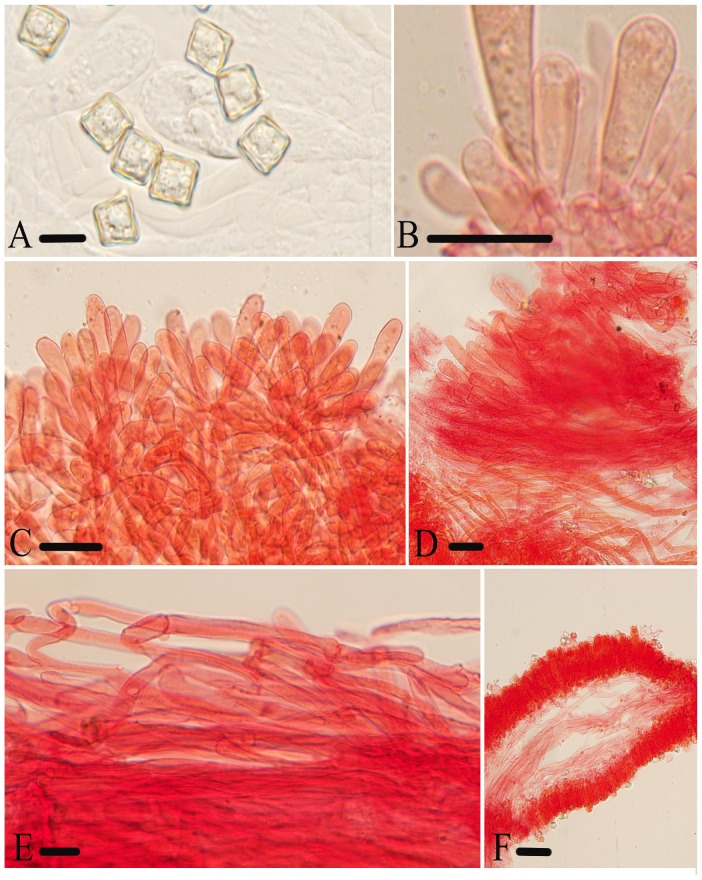
Micromorphological structures of *Entoloma submurrayi.* (**A**) Basidiospores. (**B**) Basidia. (**C**) Cheilocystidia. (**D**) Pileipellis. (**E**) Stipitipellis. (**E**) Lamellar trama. Scale bars: (**A**) 10 μm; (**B**–**F**) 30 μm. All structures were observed in 5% KOH, and 1% Congo red was used as the stain, except (**A**).

*Description*. *Basidiomata* are small, mycenoid. *Pileus* are 10–24 mm wide, campanulate, conical, or broadly conical with obvious acute papilla, translucently striate almost up to the center, smooth and glabrous, light yellow to golden yellow (3A4–5, 5AB7), lighter or fading towards rugouse margin exceeding the lamellae. *Lamellae* are moderately distant, 2.0–3.0 mm wide, with two types of lamellules, adnexed, sinuate, subventricose, concolored with cap, with a wavy or serrate concolorous edge. *Stipe* are 21–65 × 2.0–5.0 mm, central, terete, hollow, equal, white-yellow or concolorous with the pileus, with longitudinal or oblique striae, finely pruinose in the upper part elsewhere smooth and glabrous, and have a base with white tomentum. *Context* is thin, concolorous to the surface. *Odor* is indistinct, *taste* is not tested.

*Basidiospores* are (8.0)8.5–11.0 (12.0) × 8.0–10.5 (11.5) μm, (av = 9.6 × 9.2 μm), Q = 1.0–1.1 (1.2) (Q_m_ = 1.04 ± 0.03, n = 100), isodiameterical, cuboid, thick-walled, inamyloid. *Basidia* are 42–60 × 11–16 μm, clavate, 4-spored, sterigmata 7.0–11 μm long, clamped. *Pleurocystidia* are absent. *Cheilocystidia* are 47–94 × 8.0–14 μm, serrulatum-type, irregular clusters in the sterile lamellae edge, cylindric-clavate, with rounded, rarely acute apex. *Lamellar trama* are regular, made up of cylindrical hyphae 5.0–18 µm wide. *Pileipellis* is a 2-layered, suprapellis cutis with transitions to a trichoderm towards the center, made up of cylindrical hyphae 10–15 μm broad, thin-walled, slightly constricted at septa, with rounded or acute end; subpellis made up of thinner cylindrically oleiferous hyphae, up to 9.0 μm wide, all with pale yellow membranal pigment. *Stipitipellis* is a cutis composed of cylindrical hyphae 5.0–12 μm wide, with rounded or acute end. *Clamp connections* are abundant in all parts of the basidiocarp.

*Habitat*. Scattered or solitary on soil in mixed coniferous-broad-leaved forest.

Distribution. China.

*Additional specimens examined*. China, Jiangxi Province, Tongmuguan Pass of Wuyi Mountain, 27°48′54″ N, 117°43′7″ E, alt. 1130 m, 7 July 2019, collected by Jun-Qing Yan, HFJAU1050; Jiujiang City, Lushan Botanical Garden, 29°32′51″ N, 115°59′0″ E, alt. 1125 m, 9 July 2019, collected by Hong-Zhao Pan, HFJAU1062; Fujian Province, Wuyishan City, 27°42′38″ N, 117°51′30″ E, alt. 521 m, 13 August 2021, collected by Qin Na, Yu-Peng Ge, and Yu-Lan Sun, HFJAU3152. 

*Notes*. In the IL trees, *E. submurrayi* formed a sister lineage with *E. murrayi*, but the latter differs in its pileus margin not exceeding the lamellae, lack of clamp connections [38], and differences in the ITS region, with a similarity of 95%. Additionally, *E. cremeoluteum* (Largent) Noordel. and Co-David, *E. flavoquadratum* and *E. pseudomurrayi* Eyssart., Ducousso and Buyck are rather similar to the new species. Nevertheless, they exhibit distinct differences. *E. cremeoluteum* is characterized by the rostrate-ventricose cheilocystidia and pleurocystidia, and rare clamp connections [4]. *E. flavoquadratum* is characterized by its tricholomatoid basidiomata and shorter cheilocystidia (18.5–36 μm) [46]. *E. pseudomurrayi* is distinguished by the cheilocystidia of various shapes, from clavate, cylindrical, and long lanceolate to moniliform [47].

***Entoloma tomentosum*** J.Q. Yan, L.G. Chen, and S.N. Wang, sp. nov. (Figure 4E,F and Figure 11).

MycoBank: MB854071

*Etymology*. According to the features of pileus—tomentose.

*Holotype*. China, Fujian Province, Wuyishan City, Wuyi Mountain, 27°58′49″ N, 118°3′49″ E, alt. 1392 m, 17 August 2023, collected by Nian-Kai Zeng, Cheng-Feng Nie, Hua-Zhi Qin, Hui Deng, Tian Jiang, and Run-Xiang Zhao, HFJAU5159.

*Diagnosis*. *Entoloma tomentosum* is mainly characterized by the rather small basidiomata; conical to plano-convex with or without papilla, tomentose or velvety-scaly, white to pale yellow, and striate pileus; pileus margin exceeding the lamellae; adnate, lamellae; tomentose or glabrous stipe; cuboid basidiospores; heterogeneous lamella edge; versiform cheilocystidia with brown-yellow contents; and the presence of clamp connections. It differs from *E. albidoquadratum* by its smaller basidiospores, heterogeneous lamella edge, and absence of pleurocystidia.

*Description*. *Basidiomata* are rather small. *Pileus* are 12–25 mm wide, conical, campanulate to plano-convex with or without papilla, densely or sparsely tomentose or velvety-scaly, not translucently striate when young, becoming strikingly translucently striate up to the center on maturity, with straight or slightly inrolled margin, exceeding the lamellae, white to pale yellow (1–4A1 to 1–4A3), sometimes yellowish green at the center (30C8). *Lamellae* are sparse to moderately distant, 1.5–2.0 mm wide, with two types of lamellules, adnexed, ventricose, white at first, becoming pink, with wavy and concolorous edge. *Stipe* are 30–60 × 2.0–4.0 mm, central, terete, hollow, equal, white, glabrous or tomentose, and have a base with white tomentum. *Context* is thin, white. *Odor* is indistinct, *taste* is not tested.

**Figure 11 jof-10-00594-f011:**
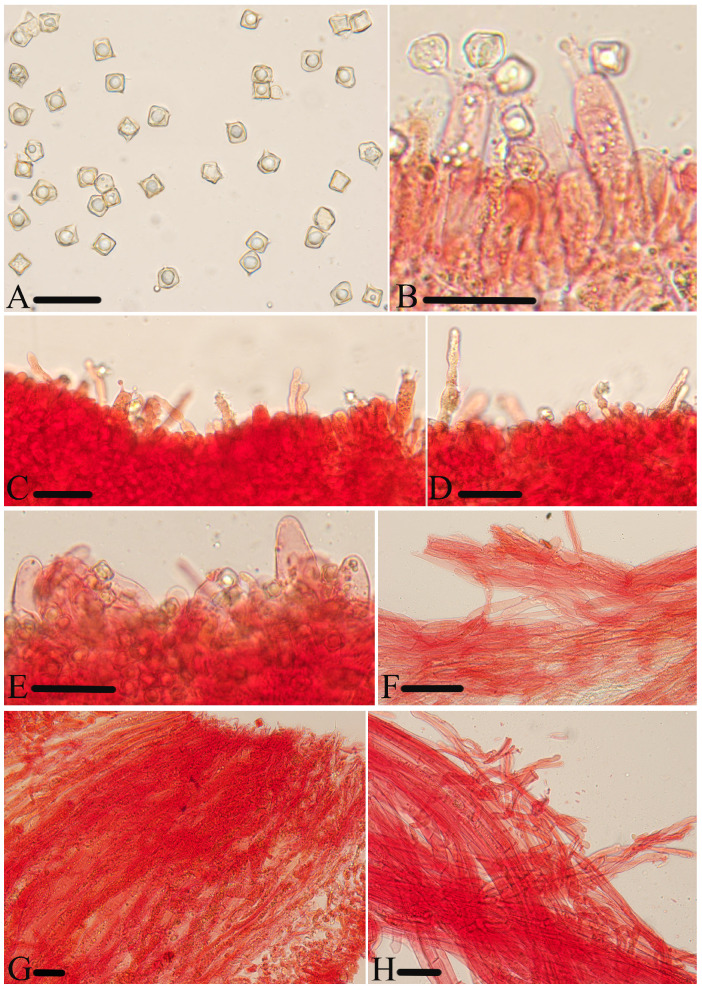
Micromorphological structures of *Entoloma tomentosum.* (**A**) Basidiospores. (**B**) Basidia. (**C**,**D**) Cheilocystidia. (**E,F**) Pileipellis. (**G**) Lamellar trama. (**H**) Stipitipellis. Scale bars: (**A**–**H**) 30 μm. All structures were observed in 5% KOH, and 1% Congo red was used as the stain, except (**A**).

*Basidiospores* are (7.5) 8.0–10.0 (11.0) × (7.0) 8.0–9.5 (10.5) μm, (av = 9.1 × 8.6 μm), Q = 1.0–1.1 (1.2) (Q_m_ = 1.06 ± 0.04, n = 100), isodiameterical, cuboid, sporadically with five angles in side-view, thick-walled, inamyloid. *Basidia* are 41–53 × 10–14 μm, clavate, 4- or 2-spored, sterigmata 7.0–13 μm long, clamped. *Pleurocystidia* are absent. *Cheilocystidia* are 35–113 × 6.0–11 μm, regularly dispersed along the heterogeneous edge, versiform, cylindrical, long lanceolate, or moniliform, with rounded or acute apex and brown-yellow contents. *Lamellar trama* are regular, made up of cylindrical hyphae 5.0–13 µm wide, intertwined by oleiferous hyphae near the margin. *Pileipellis* is a cutis with transitions to a trichoderm, made up of cylindrical hyphae 8.0–17 μm broad, thin-walled, somewhat constricted at septa, with acute end and pale yellow epiparietal pigment. *Stipitipellis* is a cutis composed of cylindrical hyphae 5.0–9.0 μm wide, even at the septa, with a rounded end. *Brilliant granules* are abundant and *clamp connections* are present in all tissue.

*Habitat*. Scattered on soil or rotten wood in mixed coniferous-broad-leaved forest.

Distribution. China.

*Additional specimens examined*. China, Fujian Province, Wuyishan City, Wuyi Mountain, 27°58′49″ N, 118°3′49″ E, alt. 1392 m, 16 August 2023, collected by Nian-Kai Zeng, Cheng-Feng Nie, Hua-Zhi Qin, Hui Deng, Tian Jiang, and Run-Xiang Zhao, HFJAU5116; 17 August 2023, HFJAU5153, HFJAU5160, HFJAU5166.

*Notes*. In the two phylogenetic trees, *E. pallidoflavum* is the closest species to the new species, but differs in the cylindrical and hyaline cheilocystidia [38], and shows only 98.4%, 96.8%, and 98.5% similarity with new species in ITS, *tef-1α*, and *rpb2* sequences. Morphologically, the new species has much in common with *E. albidoquadratum* from India, regarding the white-to-pale-yellow pileus and the versiform cheilocystidia. However, *E. albidoquadratum* is distinguished by its fimbriate and sterile lamella edge, larger basidiospores (11–16.5 × 9–14 μm), hyaline cheilocystidia, and the presence of pleurocystidia that are similar in appearance to cheilocystidia [39].

## 4. Discussion

According to the previous studies of Karstedt et al., the subgenus *Cubospora* formed a monophyly placed in the/Inocephalus-Cyanula, with the synapomorphies of mycenoid, collybioid, or tricholomatoid habit and cuboid with dihedral base basidiospores [9].

In China, about 20 species of *Entoloma* subgenus *Cubospora* were distributed in southern regions, with the white species typically identified as *E. album*, the orange species as *E. quadratum*, and the yellow species as *E. murrayi*. The new species of this study exhibit significant differences from the three aforementioned species. Additionally, some sequences that have been identified as *E. quadratum* and *E. murrayi* are noteworthy for not clustering on a single branch but forming many independent branches. This suggests that the current understanding of the distribution of these two species may not be as widespread as previously reported, and there may be some new species that are morphologically very similar to these two species which have not yet been reported.

## 5. Conclusions

Seven new species described from subtropical regions of China were well supported based on phylogenetic analysis and morphological characteristics. This study further confirmed a taxonomic relationship among the species of subgenus *Cubospora* and enhanced the species diversity of subtropical regions of China.

## 6. Key to Related Species

1. Pileus white or pale beige         2

1’ Pileus other colored            11

2. Lamellae edge heterogeneous        3

2’ Lamellae edge sterile or fertile       4

3. Pileus depressed, estriate; lamellae decurrent; basidiospores 9.0–11.5 × 7.5–10.5 μm

               E. cuboidoalbum

3’ Pileus epapillose or papillose, striate; lamellae adnexed; basidiospores 8.0–10 × 8.0–9.5 μm

                 ***E. tomentosum***

4. Clampless                 5

     

4’ Clamped                   6

5. Pileus campanulate to hemispherical with acute papilla; lamellae adnexed to adnate with serrate edge; basidiospores 7.0–9.5 μm

                   *E. album*

5’ Pileus applanate with depression; lamellae adnate to decurent with fimbriate edge; basidiospores 5.0–7.0 μm

                  *E. overeemii*

6. Basidiospores 11–16.5 × 9.0–14 μm; cheilocystidia versiform; pleurocystidia present

               E. albidoquadratum

6’ Basidiospores smaller; cheilocystidia not versiform; pleurocystidia absent

                      7

7. Pileus obviously papillate          8

7’ Pileus epapillate or unobviously papillate   9

8. Pileus not hygrophanous; lamellae adnexed; cheilocystidia carneogriseum-type; brilliant granules absent         ***E. lacticolor***

8’ Pileus hygrophanous; lamellae adnate-emarginate; cheilocystidia porphyrogriseum-type; brilliant granules abundant  E. peristerinum

9. Cheilocystidia carneogriseum-type; pileus glabrous or erected-velutinous; lamellae adnexed                ***E. subcycneum***

9’ Cheilocystidia other type           10

10. Pileus 10–25 mm, smooth and glabrous; caulocystidia present

                   E. cycneum

10’ Pileus 30–100 mm, fibrillose; caulocystidia absent

                 E. pallidoflavum

11. Pileus pink to brown            12

11’ Pileus orange to yellow              13

12. Lamellae with tiny lateral veins; pileus striate

               E. phlebophyllum

12’ Lamellae without tiny lateral veins; pileus striate

                     *E. carneum*

13. Pileus orange, conical or campanulate with or without distinct acute papilla, glabrous to fibrillose adnexed to almost free with fimbriate edge; brilliant granules abundant              *E. quadratum*

13’ Pileus yellow to brown yellow        14

14. Neither cystidia nor clamp connections *   E. phleboides*

14’ Either cystidia or clamp connections or both  15

15. Cystidia absent; basidiospores 6.0–9.0 μm *  E. gracilius*

15’ Cystidia present; basidiospores larger       16

16. Clampless; pileus innately fibrillose; lamellae adnexed to adnate

                    E. murrayi

16’ Clamped                   17

17. Lamellae edge red-brown underlined

                ***E. rufomarginatum***


17’ Lamellae edge not underlined        18

18. Pileus with depressed centre, mycenoid, glabrous, estriate; lamellae sparse, adnexed; stipe smooth and glabrous    ***E. excavatum***

18’ Pileus without depressed centre        19

19. Pleurocystidia present, rostrate-ventricose; pileus broadly campanulate, hygrophanous; clamp connections presence, but rare

                *E. cremeoluteum*

19’ Pleurocystidia absent           20

20. Cheilocystidia versiform, empty or filled with an emulsified or crystallized yellow pigment      *E. pseudomurrayi*

20’ Cheilocystidia not versiform        21

21. Pileus without papilla, convex to hemispherical, glabrous to fibrillose, striate; lamellae adnexed to almostfree with fimbriate edge  *E. luteum*

21’ Pileus with papilla              22

22. Pileus margin exceeding the lamellae     23

22’ Pileus margin not exceeding the lamellae    24

23. Pileus tricholomatoid, estriate; caulocystidia present

               E. flavoquadratum

23’ Pileus mycenoid, striate; caulocystidia absent

                 ***E. submurrayi***


24. Lamellae free; stipe fibrillose; on rotten wood

                     *E. avilanum*

24’ Lamellae adnexed; stipe glabrous; on soil   *E. plicatum*

## Figures and Tables

**Figure 1 jof-10-00594-f001:**
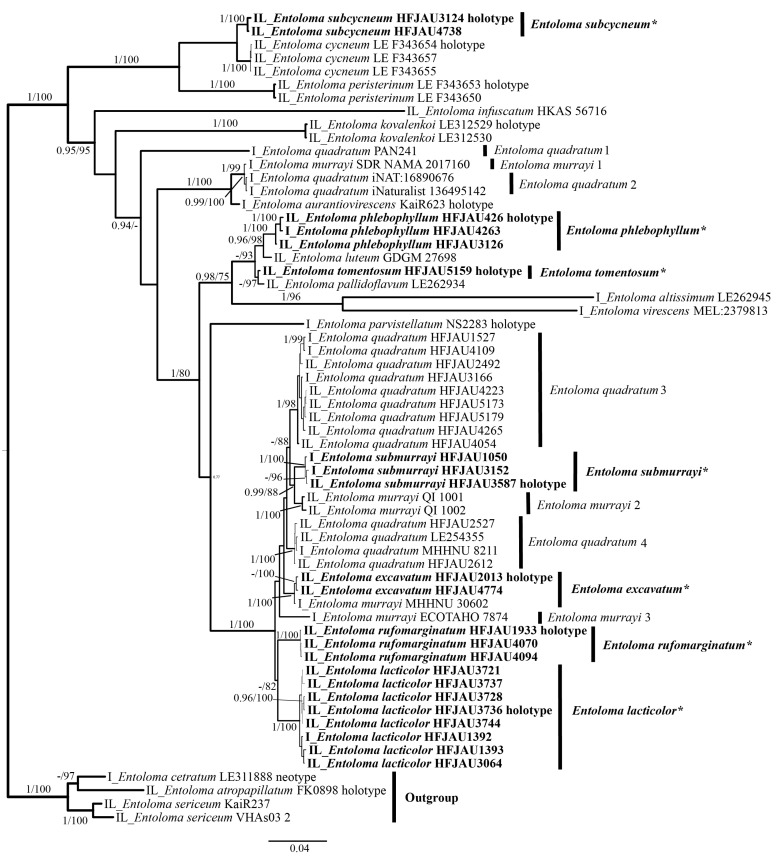
Phylogram of *Entoloma* subgenus *Cubospora* spp. generated by Bayesian inference (BI) analysis based on ITS (I) and LSU (L), rooted with *E.* subgenus *Nolanea* spp. Bayesian inference (BI-PP) ≥ 0.95 and ML bootstrap proportions (ML-BP) ≥ 75 are indicated as PP/BP. The new taxa are marked in bold and *.

**Figure 2 jof-10-00594-f002:**
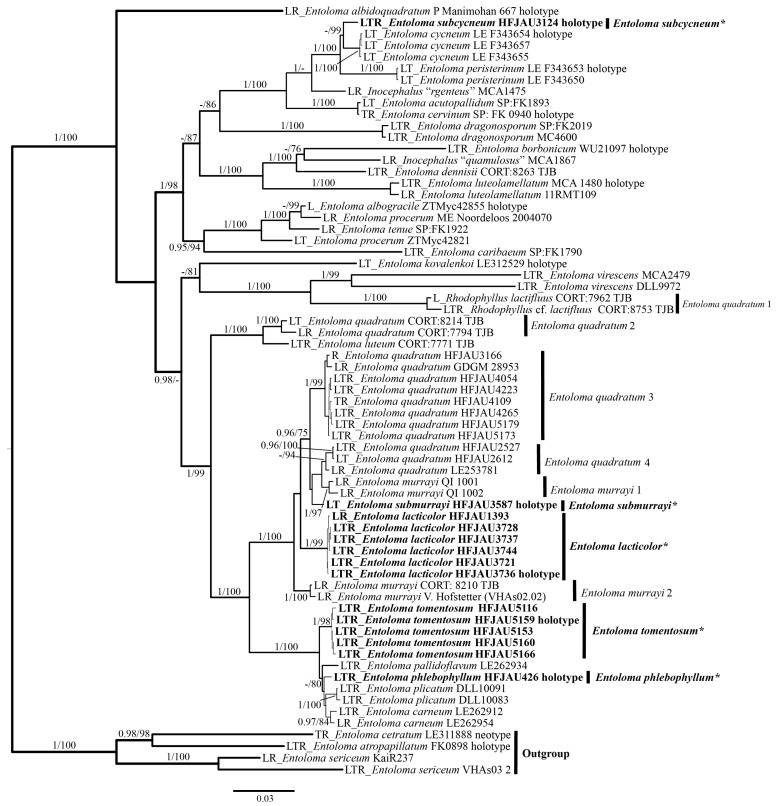
Phylogram of *Entoloma* subgenus *Cubospora* spp. generated by Bayesian inference (BI) analysis based on LSU (L), *tef-1α* (T), and *rpb2* (R), rooted with *E.* subgenus *Nolanea* spp. Bayesian inference (BI-PP) ≥ 0.95 and ML bootstrap proportions (ML-BP) ≥75 are indicated as PP/BP. The new taxa are marked in bold and *.

**Table 1 jof-10-00594-t001:** Details of sequences used in the phylogenetic analyses. Newly generated sequences are in bold.

Species	Location	Voucher Number	GenBank Number				References
			ITS	LSU	*tef-1α*	*rpb2*	
*E. acutopallidum*	Brazil	SP:FK1893	—	MG018325	MH190147	—	[9]
*E. albidoquadratum*	India	P Manimohan 667 Holotype	—	GQ289151	—	GQ289223	[2]
*E. albogracile*	Guinea	ZTMyc42855 Holotype	—	MH190207	—	—	[9]
*E. altissimum*	Vietnam	LE262945	MF476912	—	—	—	[23]
*E. atropapillatum*	Brazil	FK0898 Holotype	KF679354	KF738940	MH190137	MH190107	[9,24]
*E. aurantiovirescens*	Panama	KaiR623 Holotype	MZ611665	—	—	—	[25]
*E. borbonicum*	France:	WU21097 Holotype	—	MH190198	MH190166	MH190131	[9]
*E. caribaeum*	Brazil	SP:FK1790	—	MH190214	MH190146	MH190114	[9]
*Entoloma carneum*	Vietnam	LE262912	—	MH190181	MH190152	MH190119	[9]
*E. carneum*	Vietnam	LE262954	—	MH190184	—	MH190121	[9]
*E. cervinum*	Brazil	SP: FK 0940 Holotype	—	—	MH190138	MG018332	[9]
*E. cetratum*	Sweden	LE311888 Neotype	OL338280	—	OL405538	OL405215	[26]
*E. cycneum*	Vietnam	LE F343654 Holotype	OQ779461	OQ804518	OQ779183	—	[18]
*E. cycneum*	Vietnam	LE F343655	OQ779463	OQ804519	OQ779182	—	[18]
*E. cycneum*	Vietnam	LE F343657	OQ779464	OQ804520	OQ779184	—	[18]
*E. dennisii*	Puerto Rico	CORT:8263 TJB	—	MH190195	MH190164	MH190128	[9]
*E. dragonosporum*	Brazil	MC4600	—	MH190186	MH190156	MH190122	[9]
*E. dragonosporum*	Brazil	SP:FK2019	—	MH190179	MH190150	MG018336	[9]
** *E. excavatum* **	**China**	**HFJAU2013 Holotype**	**PP796416**	**PP789602**	—	—	**This work**
** *E. excavatum* **	**China**	**HFJAU4774**	**PP796431**	**PP789614**	—	—	**This work**
*E. infuscatum*	China	HKAS 56716	JQ281485	JQ320120	—	—	[27]
*E. kovalenkoi*	Vietnam	LE312529 Holotype	OK257210	OK257207	OK256169		[28]
*E. kovalenkoi*	Vietnam	LE312530	OK257211	OK257208	—	—	[28]
** *E. lacticolor* **	**China**	**HFJAU1392**	**OR683788**	**—**	—	—	**This work**
** *E. lacticolor* **	**China**	**HFJAU1393**	**OR683789**	**OR687487**	—	**OR738707**	**This work**
** *E. lacticolor* **	**China**	**HFJAU3064**	**OR683790**	**OR725113**	—	—	**This work**
** *E. lacticolor* **	**China**	**HFJAU3721**	**OR683791**	**OR687488**	**OR699449**	**OR738708**	**This work**
** *E. lacticolor* **	**China**	**HFJAU3728**	**OR683792**	**OR687489**	**OR699450**	**OR738709**	**This work**
** *E. lacticolor* **	**China**	**HFJAU3736 Holotype**	**OR683793**	**OR687490**	**OR699451**	**OR738710**	**This work**
** *E. lacticolor* **	**China**	**HFJAU3737**	**OR683794**	**OR687491**	**OR699452**	**OR738711**	**This work**
** *E. lacticolor* **	**China**	**HFJAU3744**	**OR683795**	**OR687492**	**OR699453**	**OR738712**	**This work**
*E. luteolamellatum*	Brazil	11RMT109	—	MH190170	—	MH190105	[9]
*E. luteolamellatum*	French	MCA 1480 Holotype	—	MH190213	MG702644	MH190135	[9]
*E. luteum*	China	GDGM 27698	JQ281486	JQ320121	—	—	[27]
*E. luteum*	USA	CORT:7771 TJB	—	MH190212	MH190161	MH190125	[9]
*E. murrayi*	China	MHHNU 30602	MK250917	—	—	—	[29]
*E. murrayi*	China	QI 1001	KJ658967	JQ993090	—	JQ993081	[30,31]
*E. murrayi*	China	QI 1002	KJ658968	JQ993089	—	JQ993082	[30,31]
*E. murrayi*	Mexico	ECOTAHO 7874	MF156254	—	—	—	Unpublished
*E. murrayi*	North American	VHAs02.02	—	GU384620	—	GU384637	[32]
*E. murrayi*	USA	CORT: 8210 TJB	—	MH190193	—	MH190127	[9]
*E. murrayi*	USA	SDR NAMA 2017160	MK575459	—	—	—	Unpublished
*E. pallidoflavum*	Vietnam	LE262934	OQ779469	MH190183	MH190155	MH259314	[9,18]
*E. parvistellatum*	Cameroon	NS2283 Holotype	MN069544	—	—	—	[33]
*E. peristerinum*	Vietnam	LE F343650	OQ779467	OQ804524	OQ779186	—	[18]
*E. peristerinum*	Vietnam	LE F343653 Holotype	OQ779466	OQ804522	OQ779188	—	[18]
** *E. phlebophyllum* **	**China**	**HFJAU3126**	**OR827451**	**OR826040**	—	—	**This work**
** *E. phlebophyllum* **	**China**	**HFJAU4261 Holotype**	**OR827447**	**OR825714**	**OR827307**	**OR827308**	**This work**
** *E. phlebophyllum* **	**China**	**HFJAU4263**	**OR827448**	—	—	—	**This work**
*E. plicatum*	Australia	DLL10083	—	JQ624612	MG702626	JQ624619	[9,34]
*E. plicatum*	Australia	DLL10091	—	JQ624613	MG702627	JQ624620	[9,34]
*E. procerum*	New Zealand	ZTMyc42821	—	MH190201	MH190167	—	[9]
*E. procerum*	Australia	ME Noordeloos 2004070	—	GQ289183	—	GQ289254	[2]
*E. quadratum*	China	GDGM 28953	—	KJ648471	—	KP226183	[31]
** *E. quadratum* **	**China**	**HFJAU1527**	**PP796414**	—	—	—	This work
** *E. quadratum* **	**China**	**HFJAU2492**	**PP796417**	**PP789603**	—	—	This work
** *E. quadratum* **	**China**	**HFJAU2527**	**PP796418**	**PP789604**	**PP873227**	**PP873244**	**This work**
** *E. quadratum* **	**China**	**HFJAU2612**	**PP796419**	**PP789605**	**PP873228**		**This work**
** *E. quadratum* **	**China**	**HFJAU3166**	**PP796422**	—	—	**PP873246**	**This work**
** *E. quadratum* **	**China**	**HFJAU4054**	**PP796425**	**PP789607**	**PP873231**	**PP873247**	**This work**
** *E. quadratum* **	**China**	**HFJAU4109**	**PP796427**	**PP789610**	**PP873232**	**PP873248**	**This work**
** *E. quadratum* **	**China**	**HFJAU4223**	**PP796428**	**PP789611**	**PP873233**	**PP873249**	**This work**
** *E. quadratum* **	**China**	**HFJAU4265**	**PP796429**	**PP789612**	**PP873234**	**PP873250**	**This work**
** *E. quadratum* **	**China**	**HFJAU5173**	**PP796437**	**PP789620**	**PP873240**	**PP873256**	**This work**
** *E. quadratum* **	**China**	**HFJAU5179**	**PP796438**	**PP789621**	**PP873241**	**PP873257**	**This work**
*E. quadratum*	China	MHHNU 8211	KU518319	—	—	—	Unpublished
*E. quadratum*	Panama	PAN241	MZ611690	—	—	—	[25]
*E. quadratum*	Russia	LE253781	—	MH190180	—	MH190118	[9]
*E. quadratum*	Russia: Far East	LE254355	KC898452	KC898504	—	—	[35]
*E. quadratum*	USA	CORT:7794 TJB	—	MH190192	—	MH190126	[9]
*E. quadratum*	USA	CORT:8214 TJB	—	MH190194	MH190162	—	[9]
*E. quadratum*	USA	iNAT:16890676	ON366783	—	—	—	Unpublished
*E. quadratum*	USA	iNaturalist 136495142	OP749675	—	—	—	Unpublished
** *E. rufomarginatum* **	**China**	**HFJAU1933 Holotype**	**PP796415**	**PP789601**	—	—	**This work**
** *E. rufomarginatum* **	**China**	**HFJAU4070**	**PP883966**	**PP789608**	—	—	**This work**
** *E. rufomarginatum* **	**China**	**HFJAU4094**	**PP796426**	**PP789609**	—	—	**This work**
*E. sericeum*	Germany	KaiR237	OL338118	OL338542	—	OL405220	[26]
*E. sericeum*		VHAs03 2	DQ367430	DQ367423	DQ367428	DQ367435	Unpublished
** *E. subcycneum* **	**China**	**HFJAU3124 Holotype**	**PP796420**	—	**PP873229**	**PP873245**	**This work**
** *E. subcycneum* **	**China**	**HFJAU4738**	**PP796430**	**PP789613**	—	—	**This work**
** *E. submurrayi* **	**China**	**HFJAU1050**	**MN622719**	—	—	—	**This work**
** *E. submurrayi* **	**China**	**HFJAU3152**	**PP796421**	—	—	—	**This work**
** *E. submurrayi* **	**China**	**HFJAU3587 Holotype**	**PP796423**	**PP789606**	**PP873230**	—	**This work**
*E. tenue*	Brazil	SP:FK1922	—	MH190176	—	MH190115	[9]
** *E. tomentosum* **	**China**	**HFJAU5116**	**PP796432**	**PP789615**	**PP873235**	**PP873251**	**This work**
** *E. tomentosum* **	**China**	**HFJAU5153**	**PP796433**	**PP789616**	**PP873236**	**PP873252**	**This work**
** *E. tomentosum* **	**China**	**HFJAU5159 Holotype**	**PP796434**	**PP789617**	**PP873237**	**PP873253**	**This work**
** *E. tomentosum* **	**China**	**HFJAU5160**	**PP796435**	**PP789618**	**PP873238**	**PP873254**	**This work**
** *E. tomentosum* **	**China**	**HFJAU5166**	**PP796436**	**PP789619**	**PP873239**	**PP873255**	**This work**
*E. virescens*	Australia	DLL9972	—	KR869937	MG702628	KR869957	[9,36]
*E. virescens*	Guyana	MCA2479	—	GU384622	MG702629	GU384640	[9,32]
*E. virescens*		MEL:2379813	MF977981	—	—	—	Unpublished
*Inocephalus “argenteus”*		MCA1475	—	GU384619	—	GU384636	[32]
*I. “squamulosus”*		MCA1867	—	GU384621	—	GU384638	[32]
*Rhodophyllus* cf. *lactifluus*		CORT:8753 TJB	—	MH190196	MH190165	MH190129	[9]
*Rh. lactifluus*		CORT:7962 TJB	—	AF261304	—	—	[37]

## Data Availability

All alignments for phylogenetic analyses were deposited in TreeBASE (http://www.treebase.org); the following links were available: http://purl.org/phylo/treebase/phylows/study/TB2:S31496?x-access-code=a3f46a59482fe0426a9347d545e5ba6e&format=html (accessed on 18 August 2024)).

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
