# Peer review of "Seven New Species of Entoloma Subgenus Cubospora (Entolomataceae, Agaricales) from Subtropical Regions of China"

_jof, 2024, doi:10.3390/jof10080594_

Round 1

Reviewer 1 Report

The authors raised the important problem of delimiting morphologically similar species in the subgenus Cubospora of the genus Entoloma. Unfortunately, they did not do this carefully enough, and the problem still remained. The key is too formal, more detail must be added. However, the work serves the task of accumulating data on this group, and therefore can be published after eliminating a number of comments.

In the methodological part, it is necessary to determine the criteria by which new species are identified on the basis of phylogeny. What distance between sequences indicates that this is a formed new species.

In is not clear what work did the authors rely on when using terminology to describe species, for example, types of cheilocystidia.

The sources of origin of the sequences (references) are not indicated.

Comments to species:

Entoloma excavatum.

Holotype must be designated. According to the tree the specimen E. murrayi MHHNU 30602 is also this new species. This should be discussed. The specimen data must be added to the distribution. In some other cases – as well.

Entoloma lacticolor

Holotype must be designated. Add comparison with E. subcycneum

Entoloma phlebophyllum

Holotype must be designated.

It should be noted that the comparison of sequences is not with the type specimens, but with the specimens identified as these species, possibly erroneously. Tiny lateral veins on sides of lamellae only - unreliable feature for distinguishing species, can depend on growing conditions, I recommend to look at what other differences from E. carneum there are.

Entoloma rufomarginatum

Holotype must be designated. Some references are missed. It is necessary to explain more clearly what the red-brown line is on the lamellae, because cheilocystidia are not colored. What is its origin?

Entoloma subcycneum

Holotype must be designated.

The exact percentage of similarity with Entoloma cycneum is not specified. In terms of morphology, the absence of brilliant granules in E. subcycneum is very strange. Most likely they are overlooked, since this is a common feature of representatives of Cubospora. Most likely they are present in E. pallideflavum also because there is a latex. E. cycneum and E. subcycneum are very close species. Perhaps it is better to describe not a new species, but a variety.

Entoloma submurrayi

Holotype must be designated. Some references are missed.

Entoloma tomentosus

Holotype must be designated. The genus Entoloma in Latin is neuter. Should be Entoloma tomentosum.

Details are in the text attached.

Author Response

Comments 1: The authors raised the important problem of delimiting morphologically similar species in the subgenus Cubospora of the genus Entoloma. Unfortunately, they did not do this carefully enough, and the problem still remained. The key is too formal, more detail must be added. However, the work serves the task of accumulating data on this group, and therefore can be published after eliminating a number of comments.

Response 1: Thank you for pointing this out. We agree with this comment. Therefore, the key has beenbe revised, adding more detail. These changes can be found – pages 25–27, lines 543, 546, 559, 569, 571, 581, 583, 590, 594, 596, 599, 602.

Comments 2: In the methodological part, it is necessary to determine the criteria by which new species are identified on the basis of phylogeny. What distance between sequences indicates that this is a formed new species.

Response 2: Thank you for pointing this out. The identifying criteria for new species are according to the viewpoints proposed by Dettman et al. 2003. These changes can be found – page 3, lines 101–102.

Comments 3: In is not clear what work did the authors rely on when using terminology to describe species, for example, types of cheilocystidia.

Response 3: Thank you for your valuable comment regarding the references of describe species. The morphological description is based on the work of Noordeloos et al. 2022. These changes can be found– page 2, lines 68–69.

Comments 4: The sources of origin of the sequences (references) are not indicated.

Response 4: Thank you for your valuable comment regarding the sources of origin of the sequences (references). The references have been added in Table 1. These changes can be found– pags 3–5, lines 103–106.

Comments 5: Entoloma excavatum. Holotype must be designated. According to the tree the specimen E. murrayi MHHNU 30602 is also this new species. This should be discussed. The specimen data must be added to the distribution. In some other cases – as well.

Response 5: Thank you for your valuable comment. We agree with this comment. Therefore, the holotype of E. excavatum has been designated and the specimen E. murrayi MHHNU 30602 has been discussed. These changes can be found – pages 10–12, lines 153, 191–193.

Comments 6: Entoloma lacticolor. Holotype must be designated. Add comparison with E. subcycneum

Response 6: Thank you for your valuable comment. We agree with this comment. Therefore, the holotype of E. lacticolor has been designated and the comparison with E. subcycneum has been added. These changes can be found – pages 13–15, lines 206, 252–252.

Comments 7: Entoloma phlebophyllum. Holotype must be designated. It should be noted that the comparison of sequences is not with the type specimens, but with the specimens identified as these species, possibly erroneously. Tiny lateral veins on sides of lamellae only - unreliable feature for distinguishing species, can depend on growing conditions, I recommend to look at what other differences from E. carneum there are.

Response 7: Thank you for your valuable comment. We agree with this comment. Therefore, the holotype of E. phlebophyllum has been designated. Although these specimens of comparison are not type specimens, some of them have referred to type specimens like E. carneum (Morozova et al. 2012) during the identification process or are paratype specimens like E. plicatum (Largent et al. 2013), so we consider the identification results of these specimens to be highly reliable. Other differences between E. phlebophyllum and E. carneum apart from tiny lateral veins have been added, as E. phlebophyllum has an estriate cap while E. carneum has a distinctly striate pileus. These changes can be found – pages 16–17, lines 263, 269–270, 302.

Comments 8: Entoloma rufomarginatum. Holotype must be designated. Some references are missed. It is necessary to explain more clearly what the red-brown line is on the lamellae, because cheilocystidia are not colored. What is its origin?

Response 8: Thank you for your valuable comment. We agree with this comment. Therefore, the holotype of E. rufomarginatum has been designated. Although we have not found the reference to protologue of E. kamerunense, Horak's identification was referred to the type specimen, thus lending credibility to the results. The red-brown tissue were not observed in H2O on dried specimens, we suppose that it may be observed in fresh specimens. These changes can be found – page 18, line 318.

Comments 9: Entoloma subcycneum. Holotype must be designated. The exact percentage of similarity with Entoloma cycneum is not specified. In terms of morphology, the absence of brilliant granules in E. subcycneum is very strange. Most likely they are overlooked, since this is a common feature of representatives of Cubospora. Most likely they are present in E. pallideflavum also because there is a latex. E. cycneum and E. subcycneum are very close species. Perhaps it is better to describe not a new species, but a variety.

Response 9: Thank you for your valuable comment. The holotype of E. subcycneum has been designated and the exact percentage of similarity with E. cycneum has been presented in manusript. We have observed multiple basidiomata of E. cycneum, yet brilliant granules still were absent. There are significant differences between E. cycneum and E. subcycneum in ITS region, with a similarity of only 95.4%, and form two stable branches in IL tree, thus it is concluded that they represent two distinct species. These changes can be found – pages 20–21, lines 370, 379, 406–408.

Comments 10: Entoloma submurrayi. Holotype must be designated. Some references are missed.

Response 10: Thank you for your valuable comment. We agree with this comment. Therefore, the holotype of E. submurrayi has been designated and the reference of E. flavoquadratum has been added. These changes can be found – pages 2122, lines 421, 469470.

Comments 11: Entoloma tomentosus. Holotype must be designated. The genus Entoloma in Latin is neuter. Should be Entoloma tomentosum.

Response 11: Thank you for your valuable comment. We agree with this comment. Therefore, the holotype of E. tomentosum has been designated and the name “E. tomentosus” has been renamed to “E. tomentosum”. These changes can be found – page23, lines 473, 476.

Reviewer 2 Report

It is a great contribution on the genus Entoloma. Only is necessary to improve some points.

Lines 33-35. In these lines, please insert additional cites, e.g. Kirk et al. 2008, Kalichman et al. 2021 He et al. 2019, 2024 regarding to the number of species in Entoloma.

Lines 43-36. How is the relationship between these two clades Cubospora and Cuboeccilia… are sister clade or are separate clades.

Lines 49-52. Please write it as aim or as hypothesis, not as Results

Lines 520-528. Please improve the discussion focused in the monophyly of the Cubospora group an and synapomorphies of the clade

Lines 530-531. The first sentence is not a conclusion is a result.

Author Response

Comments 1: Lines 33-35. In these lines, please insert additional cites, e.g. Kirk et al. 2008, Kalichman et al. 2021 He et al. 2019, 2024 regarding to the number of species in Entoloma.

Response 1: Thank you for your valuable comment. We agree with this comment. Therefore, we have cited He et al. 2019 and Asif et al. 2024. This change can be found – page 1, lines 3435.

Comments 2: Lines 43-46. How is the relationship between these two clades Cubospora and Cuboeccilia… are sister clade or are separate clades.

Response 2: Thank you for your valuable comment. We agree with this comment. The relationship between these two clades Cubospora and Cuboeccilia are separate clades. This change can be found – page 1, lines 4346.

Comments 3: Lines 49-52. Please write it as aim or as hypothesis, not as Results

Response 3: Thank you for your valuable comment. We agree with this comment. Therefore, we have revised it as aim. This change can be found – page 2, lines 4952.

Comments 4: Lines 520-528. Please improve the discussion focused in the monophyly of the Cubospora group and synapomorphies of the clade

Response 4: Thank you for your valuable comment. We agree with this comment. Therefore, we have demonstrated the monophyly of the Cubospora group and enumerated some synapomorphies. This change can be found – page 25, lines 525527.

Comments 5: Lines 530-531. The first sentence is not a conclusion is a result.

Response 5: Thank you for your valuable comment. After careful consideration, we considered that this sentence is both a conclusion and a result.

Round 2

Reviewer 1 Report

The work contains new important information about new species of the genus Entoloma, and will undoubtedly be in demand by scientists studying this genus or the general diversity of Asian fungi.

In the abstract, it is needed to add from which species the new species differ in the indicated characteristics, and also to make additions according to the changes in the text of the article.

103-104 It is advisable not only to provide a reference to the work, but also to list the criteria used. What differences between species indicate that this is truly a new species, and not a form or variety?

212 – without a dot

230 (Entoloma lacticolor) – carneogriseum-type. It cannot be of poliopus-type, since it obviously has a tramal origin. The shape of the end cells is not club-shaped, but sub-club-shaped or cylindrical.

377 –(Entoloma subcycneum) – usually brilliant granules abundant if oleiferous hyphae present

507 - not all hyphae of trama are oleiferous. Not made up... But oleiferous hyphae are included...

565 – usually brilliant granules abundant if oleiferous hyphae present

574 – absent

587 - absent

In the microstructure drawings, the red color is so saturated that the boundaries between the cells are not visible. It is necessary to mute it

Author Response

Comments 1: The work contains new important information about new species of the genus Entoloma, and will undoubtedly be in demand by scientists studying this genus or the general diversity of Asian fungi.

Response 1: Thank you for your valuable comment.

Comments 2: In the abstract, it is needed to add from which species the new species differ in the indicated characteristics, and also to make additions according to the changes in the text of the article.

Response 2: Thank you for pointing this out. We have carefully verified the mainly identifying characteristics of new species referring to the morphological differences from similar species mentioned in Diagnosis. If such from which species the new species differ in the indicated characteristics is added, it would not only inevitably overlap with the content of the Diagnosis but also render the abstract overly verbose and redundant. Therefore, after careful consideration, we have decided not to add this from in abstract

Comments 3: 103-104 It is advisable not only to provide a reference to the work, but also to list the criteria used. What differences between species indicate that this is truly a new species, and not a form or variety?

Response 3: Thank you for your valuable comment. We agree with this comment. The criteria has been listed in manuscript, exhibiting 1 to 2 stable morphological differences from similar species and forming separated and stable clades in phylogenetic tree. The change can be found–lines 103–104.

Comments 4: 212 – without a dot

Response 4: Thank you for your valuable comment. We agree with this comment. Therefore, the dot has been deleted. The change can be found– lines 214.

Comments 5: 230 (Entoloma lacticolor) – carneogriseum-type. It cannot be of poliopus-type, since it obviously has a tramal origin. The shape of the end cells is not club-shaped, but sub-club-shaped or cylindrical.

Response 5: Thank you for your valuable comment. We agree with this comment. Therefore, the type of cheilocystidia has been revised as carneogriseum-type and the shape has been revised as cylindrical to subclavate. These changes can be found – lines 232–233.

Comments 6: 377 –(Entoloma subcycneum) – usually brilliant granules abundant if oleiferous hyphae present

Response 6: Thank you for your valuable comment. We agree with this comment. In E. subcycneum, due to the lack of clarity in the initial description, upon further confirmation, it was discovered that these hyphae near the margin of lamellar trama lacked oil droplets within their interiors and should be refractive hyphae, therefore, the brilliant granules are indeed absent. The change can be found – lines 394.

Comments 7: 507 - not all hyphae of trama are oleiferous. Not made up... But oleiferous hyphae are included...

Response 7: Thank you for your valuable comment. We agree with this comment. We have revised the inaccurate phrasing and modified it to made up of cylindrical hyphae..., intertwined by oleiferous hyphae....The change can be found – lines 506–507.

Comments 8: 565 – usually brilliant granules abundant if oleiferous hyphae present

Response 8: Thank you for your valuable comment. We agree with this comment. In E. lacticolor, due to the lack of clarity in the initial description, upon further confirmation, it was discovered that these hyphae near the margin of lamellar trama lacked oil droplets within their interiors and should be refractive hyphae, therefore, the brilliant granules are indeed absent. The change can be found – lines 235.

Comments 9: 574 – absent

Response 9: Thank you for your valuable comment. We agree with this comment. The word has been corrected. The change can be found – lines 575.

Comments 10: 587 - absent

Response 10: Thank you for your valuable comment. We agree with this comment. The word has been corrected. The change can be found – lines 588.

Comments 11: In the microstructure drawings, the red color is so saturated that the boundaries between the cells are not visible. It is necessary to mute it

Response 11: Thank you for your valuable comment. We agree with this comment. Therefore, the Figures 5E, 7J, 11G have been replaced with clearer ones.

Comments 12: It is recommended to check for cheilocystidia types in all new species and the presence of brillant granules. All cheilocystidia presented are of the tramal origin - serrulatum-type, porphyrogriseum-type or carneogriseum-type, depending on the type of arranging (in clasters or trichodermally) and on the form of the terminal cells (subclavate, cylindrical or tapering).

Response 12: Thank you for your valuable comment. We agree with this comment. We have check for cheilocystidia types in all new species and the presence of brillant granules. The cheilocystidia type of E. lacticolor have been corrected to carneogriseum-type and find that brillant granules are present in only E. phlebophyllum and E. tomentosum. These changes can be found – lines 294, 510.
